# GEOMETRY-AWARE 4D VIDEO GENERATION FOR ROBOT MANIPULATION

**Zeyi Liu**[1]* **Shuang Li**[1] **Eric Cousineau**[2] **Siyuan Feng**[2]
**Benjamin Burchfiel**[2] **Shuran Song**[1]

[1] Stanford University [2] Toyota Research Institute

`https://robot4dgen.github.io/`

## ABSTRACT

Understanding and predicting dynamics of the physical world can enhance a robot's ability to plan and interact effectively in complex environments. While recent video generation models have shown strong potential in modeling dynamic scenes, generating videos that are both temporally coherent and geometrically consistent across camera views remains a significant challenge. To address this, we propose a 4D video generation model that enforces multi-view 3D consistency of generated videos by supervising the model with cross-view pointmap alignment during training. Through this geometric supervision, the model learns a shared 3D scene representation, enabling it to generate spatio-temporally aligned future video sequences from novel viewpoints given a single RGB-D image per view, and without relying on camera poses as input. Compared to existing baselines, our method produces more visually stable and spatially aligned predictions across multiple simulated and real-world robotic datasets. We further show that the predicted 4D videos can be used to recover robot end-effector trajectories using an off-the-shelf 6DoF pose tracker, yielding robot manipulation policies that generalize well to novel camera viewpoints.

## 1 INTRODUCTION

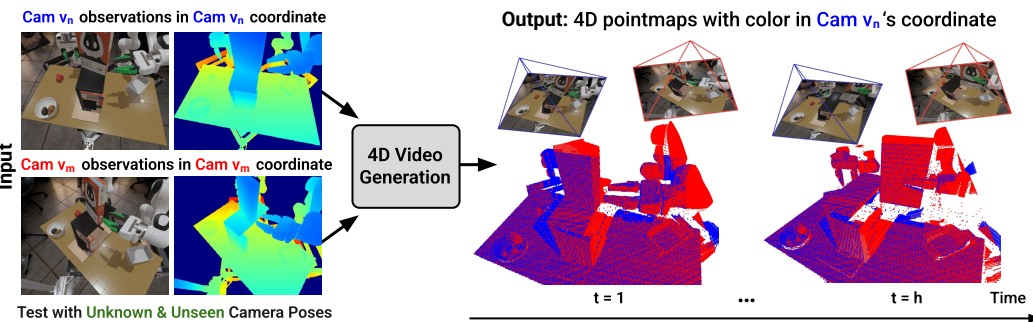

Figure 1: **Geometry-aware 4D Video Generation.** Our model takes RGB-D observations from two camera views and predicts future 4D pointmaps in the coordinate frame of the reference view $v_n$. The blue pointmap is predicted from camera $v_n$, while the red pointmap shows the prediction from camera $v_m$ projected into the coordinate frame of $v_n$. RGB videos are predicted separately for each view. Together, the model enables geometry-consistent 4D video generation.

Understanding how the visual world changes with interactions is a key capability for intelligent robotic systems. For robot manipulation tasks, the robots need to anticipate how the environment changes by taking into account object motions or occlusions upon interactions over extended time

---

*work partially done at internship

horizons. Recent advancements in video generation models offer a powerful paradigm for learning such dynamic visual models: by forecasting future observations, robots can simulate possible outcomes, plan actions, and adapt to new scenarios.

However, it remains a challenge to generate realistic and physically plausible videos from which smooth and precise robot policies can be extracted from. There are two challenges: **temporal coherence**, ensuring smooth, causally consistent motion over time; and **3D consistency**, preserving object geometry and spatial correspondences across different viewpoints. Most existing video generation models capture one at the expense of the other. Pixel-based models (Yan et al., 2021; Ho et al., 2022) trained on RGB videos often excel at short-term motion but lack an understanding of 3D structures, which leads to artifacts like flickering, deformation, or object disappearance. In contrast, 3D-aware approaches enforce geometric constraints but are limited to simple, static backgrounds and struggle to scale to realistic, multi-object manipulation scenarios (Xie et al., 2024; Li et al., 2024a; Zhang et al., 2024).

In this work, we present a video generation framework that bridges this gap by unifying strong temporal modeling with robust 3D geometric consistency. Our method produces *4D videos* that can be rendered into RGB-D sequences that are both coherent over time and spatially consistent across camera views. To achieve this, we introduce a **geometry-consistent supervision** mechanism inspired by DUSt3R (Wang et al., 2024b) and adapt it for the video generation task. Specifically, the model is trained to predict a pair of 3D pointmap sequences: one for a reference view and one for a second view projected into the reference view camera coordinate frame. By minimizing the difference between reference and projected 3D points over time, the model learns a shared scene representation across views. This enables robust generalization to novel viewpoints at inference, which is particularly useful in robotic applications where even small camera view shifts can push visuomotor policies out of distribution and lead to failures.

Pretrained video diffusion models provide strong visual and motion priors learned from large-scale video datasets. To enhance temporal coherence, we initialize our model with pretrained weights and extend it to jointly generate future RGB frames and pointmaps. The RGB frames are trained using the original video generation loss, while the pointmaps are supervised using the proposed geometry-consistent loss. This combination enables the model to leverage the temporal priors of pretrained models while enforcing spatial and cross-view consistency through pointmap alignment, resulting in spatio-temporally consistent RGB-D video generation. We evaluate the 4D video generation quality on both simulated and real-world tasks, and our approach outperforms baselines in both video quality and cross-view consistency.

We further demonstrate that the predicted multi-view RGB-D videos can be directly used to extract robot end effector trajectories using an off-the-shelf 6DoF pose tracker, such as FoundationPose (Wen et al., 2024). We evaluate this approach on three simulated robot manipulation tasks where the camera views are unseen during training, achieving good success rates on all tasks and outperforms the baseline method. Additionally, our model generates future pointmaps in the reference view's coordinate frame solely based on the initial RGB-D observation in each view, without camera poses as inputs, bypassing extrinsics calibration for novel camera poses.

In summary, we propose a 4D video generation framework that achieves both 3D geometric consistency and temporal coherence. To achieve this, we first introduce a geometry-consistent supervision mechanism that enforces cross-view alignment of generated videos over time. Second, we develop a benchmark for video generation in robotic manipulation, comprising both simulation and real-world tasks. Each task is recorded from diverse camera viewpoints, enabling comprehensive evaluation of 4D generation quality and generalization to unseen views. Finally, we demonstrate that the generated 4D videos can be directly used to extract robot trajectories using an off-the-shelf 6DoF pose tracker, enabling applications to manipulation tasks.

## 2  RELATED WORK

**Video Generation** has been a long-standing task in computer vision. Early works use recurrent networks (Srivastava et al., 2015; Chiappa et al., 2017) or generative adversarial networks (Vondrick et al., 2016) to learn temporal dynamics from video data. With the recent success of image diffusion models, many works have extended diffusion models for the video prediction task by adding additional temporal layers (Yan et al., 2021; Ho et al., 2022) and using latent diffusion techniques (Blattmann

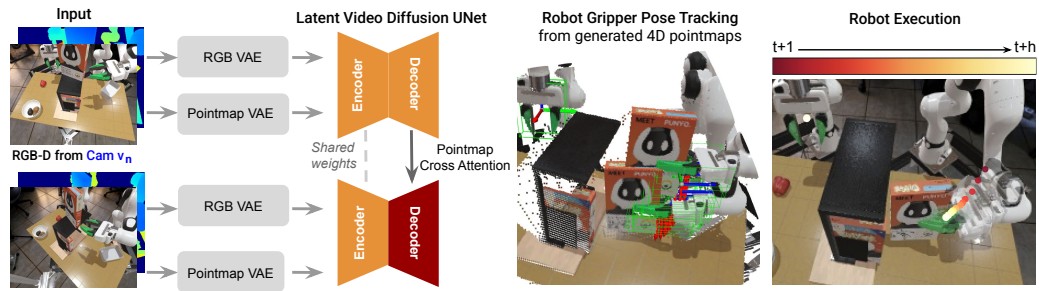

Figure 2: **4D Video Generation for Robot Manipulation.** Our model takes RGB-D observations from two camera views, and predicts future pointmaps and RGB videos. To ensure cross-view consistency, we apply cross-attention in the U-Net decoders for pointmap prediction. The resulting 4D video can be used to extract the 6DoF pose of the robot end-effector using pose tracking methods, enabling downstream manipulation tasks.

et al., 2023; Liu et al., 2024; Zheng et al., 2024; Yang et al., 2024b). In addition to RGB videos, video diffusion models have also been shown to generate other modalities such as high-fidelity depth videos (Saxena et al., 2023; Hu et al., 2024; Shao et al., 2024) and pointmaps (Nguyen et al., 2025). In this work, we adapt a latent video diffusion model (SVD) to jointly learn from RGB and depth videos with improved temporal and spatial consistency.

**Multi-View and 4D Video Generation.** Recent advances in camera-conditioned video generation improve spatial consistency by associating camera poses with multi-view videos (Van Hoorick et al., 2024; He et al., 2024; Kuang et al., 2024; Cheong et al., 2024). However, 4D video generation requires joint reasoning about both object geometry across views and motion over time. Prior 4D generation works separately optimize for temporal consistency with a video model and spatial consistency with a novel view synthesis model (Sun et al., 2024; Wang et al., 2024c; Yang et al., 2024a). In this work, we present a 4D video generation framework unifies the two objectives. In particular, our method introduces a geometry-consistent supervision mechanism through cross-view pointmap alignment, inspired by DUSt3R (Wang et al., 2024b), on top of the standard video diffusion process. This approach enforces spatial consistency across camera views while ensuring temporal coherence. Our goal aligns with recent 4D generation works that jointly optimize spatial and temporal consistency (Xie et al., 2024; Li et al., 2024a), but while they primarily focus on single-object videos with white backgrounds, we target multi-object, dynamic robot manipulation scenes.

**Generative Models for Robot Planning.** With the recent success of video generation models and their strong generalization capabilities across diverse visual scenes, many works have explored their potential as dynamics models for robotic tasks (Yang et al., 2025b). Specially, robot actions can be extracted from predicted future frames using a learned inverse dynamics model (Du et al., 2023; Tian et al., 2024; Zhen et al., 2025), a behavior cloning policy conditioned on generated outputs (Xu et al., 2024; Bharadhwaj et al., 2024; Huang et al., 2025a), or an RGB-based pose tracking model (Liang et al., 2024). To more tightly couple future state prediction with action inference, recent methods have proposed unified models that simultaneously predict both future video frames and robot actions (Zhu et al., 2025; Li et al., 2025). Yet it remains a challenge to generate spatially consistent predictions across views and capture accurate 3D geometry needed for precise robot manipulation. To bridge this gap, we propose a model that learns spatial correspondences across camera views, enabling better RGBD video generation quality on novel views and improved end effector pose tracking accuracy for better robot task performance.

## 3 METHOD

In this section, we present a 4D video generation algorithm that unifies temporal coherence with 3D geometric consistency, enabling spatio-temporally consistent predictions over time and across varying viewpoints.

**Problem Statement.** Traditional video generation models predict future RGB frames $\{\mathbf{O}_{t+1}, \cdots, \mathbf{O}_{t+h}\}$ from past observations $\{\mathbf{O}_{t-h'+1}, \cdots, \mathbf{O}_t\}$ captured from a single view, where $h$ and $h'$ are the history and future horizon respectively. However, single-view prediction lacks geometric grounding and often results in temporally plausible but spatially inconsistent outputs. To

address this, we introduce a stereo-based video generation network with additional 3D supervision to enforce consistent scene geometry across time and views. For a pair of video frames $\mathbf{O}_t^m$ and $\mathbf{O}_t^n$ taken at time $t$ from camera views $v_m$ and $v_n$, each pixel coordinate $(i, j) \in \{1, \cdots, W\} \times \{1, \cdots, H\}$ corresponds to a 3D point computed from its depth. This results in pointmaps $\mathbf{X}_t^m$ and $\mathbf{X}_t^n$, where $\mathbf{X}_t^n \in \mathbb{R}^{W \times H \times 3}$ represents the per-pixel 3D coordinates for view $v_n$, and similarly for $\mathbf{X}_t^m$. Here, $W$ and $H$ denote the image width and height, respectively.

We first describe our video diffusion backbone for generating RGB videos (§ 3.1), next we introduce our geometry-consistent supervision mechanism on pointmaps across views to enforce spatial alignment (§ 3.2). We integrate temporal dynamics with 3D geometric consistency through joint optimization (§ 3.3). Finally, we demonstrate how 4D video predictions can be used to recover robot end effector poses $\mathbf{T}_t \in \mathrm{SE}(3)$ at each time step using off-the-shelf trackers (§ 3.4).

## 3.1 DIFFUSION-BASED VIDEO GENERATION

We adopt the Stable Video Diffusion (Blattmann et al., 2023) framework which has demonstrated strong performance in generating short, temporally coherent video sequences. It first projects historical video frames $\{\mathbf{O}_{t-h+1}, \cdots, \mathbf{O}_t\}$ into a latent space using a pretrained Variational Autoencoder (VAE) (Kingma et al., 2013) encoder. A diffusion model, implemented as a U-Net with an encoder-decoder structure, then predicts future latent representations $\{\mathbf{z}_{t+1}, \cdots, \mathbf{z}_{t+h}\}$, which are decoded back into RGB frames $\{\mathbf{O}_{t+1}, \cdots, \mathbf{O}_{t+h}\}$ using VAE decoder. The diffusion model $f_\theta$ is trained using an alternative of the standard DDPM (Ho et al., 2020) method, which directly predicts the original clean data from the noisy input at each diffusion step. The training objective for predicting a future latent $z_{t'}$ at timestep $t'$ is to minimize:

$$\mathcal{L}_{\mathrm{diff}}(t') = \mathbb{E}_{\epsilon_{t'}, \mathbf{z}_{t'}(0), k} \left[ \left\| \mathbf{z}_{t'}(0) - f_\theta(\mathbf{z}_{t'}(k), k) \right\|^2 \right]$$
$$\text{where } \mathbf{z}_{t'}(k) = \sqrt{\alpha_k}\, \mathbf{z}_{t'}(0) + \sqrt{1 - \alpha_k}\, \epsilon_{t'}, \quad \epsilon_{t'} \sim \mathcal{N}(0, I) \tag{1}$$

Here $\epsilon_{t'}$ denotes Gaussian noise, $\mathbf{z}_{t'}(0)$ denotes the un-noised latent, and $\mathbf{z}_{t'}(k)$ is the noised latent at diffusion step $k$. During inference, videos are generated by progressively denoising random Gaussian noise using the trained diffusion model.

## 3.2 GEOMETRY-CONSISTENT SUPERVISION

To enforce 3D consistency across views, we adopt the cross-view pointmap supervision strategy from DUSt3R (Wang et al., 2024b), adapted to the video generation setting. As shown in Figure 2, given the history pointmaps $\{\mathbf{X}_{t-h+1}^n, \cdots, \mathbf{X}_t^n\}$ from camera view $v_n$, we first encode them using a Pointmap VAE, which is initialized from the pretrained RGB VAE from SVD (Blattmann et al., 2023) and fine-tuned on pointmap data. This produces the latent representation $\{\mathbf{z}_{t+1}^n, \cdots, \mathbf{z}_{t+h}^n\}$. We then apply the same latent diffusion method used for RGB video prediction to forecast future pointmaps in the latent space. The predicted latents are subsequently decoded by the Pointmap VAE decoder to obtain future pointmaps $\{\mathbf{X}_{t+1}^n, \ldots, \mathbf{X}_{t+h}^n\}$.

In parallel, the model also predicts future pointmaps from a second camera view $v_m$, but instead of generating them in their native frame, it expresses them in the coordinate frame of view $v_n$. This results in a sequence of projected pointmaps $\{\mathbf{X}_{t+1}^{m \to n}, \ldots, \mathbf{X}_{t+h}^{m \to n}\}$. Each of these predictions are encoded into latent representations that are aligned with view $v_n$, enabling supervision through cross-view consistency.

During training, we supervise the model $f_\theta$ at each future time step $t'$ using diffusion losses applied to both the native view $v_n$ and the projected view $v_m \to v_n$:

$$\mathcal{L}_{\mathrm{3D\text{-}diff}}(t') = \mathbb{E}_{\epsilon_{t'}^n, \mathbf{z}_{t'}^n(0), k} \left[ \left\| \mathbf{z}_{t'}^n(0) - f_\theta\left(\mathbf{z}_{t'}^n(k), k, c^n\right) \right\|^2 \right]$$
$$+ \mathbb{E}_{\epsilon_{t'}^m, \mathbf{z}_{t'}^{m \to n}(0), k} \left[ \left\| \mathbf{z}_{t'}^{m \to n}(0) - f_\theta\left(\mathbf{z}_{t'}^{m \to n}(k), k, c^m\right) \right\|^2 \right] \tag{2}$$
$$\text{where} \quad \mathbf{z}_{t'}^n(k) = \sqrt{\alpha_k}\, \mathbf{z}_{t'}^n(0) + \sqrt{1 - \alpha_k}\, \epsilon_{t'}^n,$$
$$\mathbf{z}_{t'}^{m \to n}(k) = \sqrt{\alpha_k}\, \mathbf{z}_{t'}^{m \to n}(0) + \sqrt{1 - \alpha_k}\, \epsilon_{t'}^m, \quad \epsilon_{t'}^n, \epsilon_{t'}^m \sim \mathcal{N}(0, I)$$

where $\mathbf{z}_{t'}^n(k)$ denotes the noised latent of the pointmap from view $v_n$ at diffusion step $k$, and $\mathbf{z}_{t'}^{m \to n}(k)$ is the noised latent for the pointmap from view $v_m$ projected into the coordinate frame of view $v_n$.

Similar to Equation (1), $\epsilon_{t'}^n$ and $\epsilon_{t'}^m$ are Gaussian noise added to the pointmap latents from view $v_n$ and $v_m$, respectively. The conditioning latents $c^n$ and $c^m$ are derived from the historical pointmaps of their respective views. See Appendix for model architecture details.

While camera poses are required during training to define the projection from view $v_m$ to the coordinate frame of $v_n$, a key advantage emerges at inference. Given an observation from a novel view $v_m$, the model can directly predict pointmaps in the coordinate frame of $v_n$, eliminating the need for camera poses as inputs during testing.

**Multi-View Cross-Attention for 3D consistency**. Unlike RGB video prediction, where each view predicts future frames independently in its own coordinate system and a single shared U-Net diffusion model can be used across views, pointmap prediction requires enforcing 3D alignment across views. The native view $v_n$ predicts pointmaps in its own frame, while the second view $v_m$ predicts pointmaps projected into the coordinate frame of $v_n$. To reflect this asymmetry, we use two separate decoders in the U-Net diffusion model (with identical architecture but independent weights) and introduce cross-attention layers between the decoders to enable information transfer. Specifically, the intermediate features from the decoder of view $v_n$ are passed to the corresponding stage in the decoder of view $v_m$ through cross-attention. This allows the decoder of $v_m$ to attend to and incorporate geometric cues from $v_n$, facilitating accurate pointmap prediction in the reference coordinate frame. This mechanism enhances cross-view geometric consistency, particularly under viewpoint variations.

## 3.3 Joint Temporal and 3D Consistency Optimization

The pretrained video diffusion model provides strong temporal priors for predicting scene dynamics, while the 3D pointmap supervision enforces 3D geometric consistency across views. We leverage the pretrained video model and optimize it with both the RGB-based video diffusion loss and the pointmap-based 3D consistency loss. The full training objective is defined as the sum of losses across all predicted time steps $t' \in t+1, \ldots, t+h$ and both camera views, $v_n$ and $v_m$.

$$\mathcal{L} = \sum_{t'=t+1}^{t+h} \left[ \underbrace{\mathcal{L}_{\text{diff}}^n(t') + \mathcal{L}_{\text{diff}}^m(t')}_{\text{RGB loss}} + \lambda \cdot \underbrace{\mathcal{L}_{\text{3D-diff}}(t')}_{\text{pointmap loss}} \right], \tag{3}$$

where $\lambda$ balances the contribution of the geometric supervision. We set $\lambda = 1$ in our experiments. This joint objective encourages both temporal coherence and cross-view 3D consistency.

## 3.4 Robot Pose Estimation from 4D videos

We leverage the predicted 4D video to extract robot trajectories using an off-the-shelf 6DoF pose tracking model, FoundationPose (Wen et al., 2024). The model takes as input RGB-D frames from a single view, a binary mask of the target object in the initial frame (generated using SAM2 (Ravi et al., 2024)), camera intrinsics, and the gripper CAD model. At each pose estimation timestep, the model outputs the estimated pose of the CAD model, $\mathbf{T}_t \in \text{SE}(3)$, along with a confidence score for the prediction, and tracks the object for subsequent frames.

Pose estimation is performed independently on both views, and the result with the highest confidence score is selected. With the camera extrinsics of the reference view, the tracked poses are transformed into the global frame and used to command the robot.

To infer the gripper open/close state, we segment the left and right gripper fingers and project their pixels into 3D space based on the predicted RGB-D sequences. The distance between the centroids of the two finger point clouds is measured: if it falls below a threshold $\delta$ (0.10m for StoreCerealBoxUnderShelf, 0.06m for PutSpatulaOnTable, for 0.12m PutAppleFromBowlIntoBin), the gripper is considered closed; otherwise, it is considered open. The recovered trajectories are directly used to control the robot to execute downstream tasks.

| Method | Cross-view Consist. | RGB | | Depth | | | |
|---|---|---|---|---|---|---|---|
| | mIoU ($\uparrow$) | FVD-$n$ ($\downarrow$) | FVD-$m$ ($\downarrow$) | AbsRel-$n$ ($\downarrow$) | AbsRel-$m$ ($\downarrow$) | $\delta_1$-$n$ ($\uparrow$) | $\delta_1$-$m$ ($\uparrow$) |
| **Task 1: StoreCerealBoxUnderShelf** | | | | | | | |
| OURS | **0.70** | **411.20** | **561.43** | **0.06** | **0.11** | **0.95** | **0.92** |
| OURS w/o MV attn | 0.41 | 497.43 | 607.73 | 0.15 | 0.31 | 0.75 | 0.66 |
| 4D Gaussian (Wang et al., 2024a) | 0.39 | 1208.00 | 1094.98 | 0.20 | 0.31 | 0.74 | 0.63 |
| SVD (Blattmann et al., 2023) | – | 977.06 | 743.25 | – | – | – | – |
| SVD w/ MV attn | – | 941.73 | 653.44 | – | – | – | – |
| **Task 2: PutSpatulaOnTable** | | | | | | | |
| OURS | **0.69** | 377.68 | **257.70** | **0.03** | **0.07** | **0.98** | **0.97** |
| OURS w/o MV attn | 0.44 | 451.54 | 302.29 | 0.10 | 0.33 | 0.89 | 0.41 |
| 4D Gaussian (Wang et al., 2024a) | 0.46 | 1241.13 | 815.77 | 0.33 | 0.30 | 0.43 | 0.37 |
| SVD (Blattmann et al., 2023) | – | **370.92** | 417.56 | – | – | – | – |
| SVD w/ MV attn | – | 536.02 | 445.68 | – | – | – | – |
| **Task 3: PlaceAppleFromBowlIntoBin** | | | | | | | |
| OURS | **0.64** | **490.88** | **366.98** | **0.06** | **0.07** | **0.95** | **0.96** |
| OURS w/o MV attn | 0.26 | 597.05 | 573.73 | 0.14 | 0.49 | 0.76 | 0.30 |
| 4D Gaussian (Wang et al., 2024a) | 0.44 | 1396.10 | 1191.40 | 0.18 | 0.16 | 0.80 | 0.81 |
| SVD (Blattmann et al., 2023) | – | 659.52 | 628.01 | – | – | – | – |
| SVD w/ MV attn | – | 812.94 | 766.52 | – | – | – | – |
| **Multi-task Real World Dataset** | | | | | | | |
| OURS | **0.56** | **384.26** | **331.00** | **0.10** | **0.20** | **0.93** | **0.89** |
| OURS w/o MV attn | 0.32 | 694.23 | 601.93 | 0.14 | 0.34 | 0.90 | 0.81 |
| 4D Gaussian (Wang et al., 2024a) | 0.00 | 2102.52 | 2608.94 | 0.32 | 1.80 | 0.78 | 0.22 |
| SVD (Blattmann et al., 2023) | – | 605.10 | 648.41 | – | – | – | – |
| SVD w/ MV attn | – | 1151.05 | 931.33 | – | – | – | – |

Table 1: **Multi-view 4D Video Generation Results.** We compare our method with baselines in terms of cross-view consistency, RGB video generation quality, and depth generation quality. Our method consistently enables high-quality video and depth generation while maintaining strong cross-view consistency on both simulated and real-world datasets.

# 4 EXPERIMENTS

## 4.1 TASKS

We evaluate 4D video generation results in both the LBM simulation and real world. The LBM simulation (Institute, 2025) is a physics-based simulator built on Drake (Tedrake & the Drake Development Team, 2019) that provides realistic rendering and demonstrations for robot manipulation tasks.

*Simulation Task:* We evaluate on three table-top manipulation tasks with both object and motion diversity (Figure 6): *StoreCerealBoxUnderShelf*, *PutSpatulaOnTable*, and *PlaceAppleFromBowlIntoBin*. In *StoreCerealBoxUnderShelf*, a robot arm picks up a cereal box from the top of the shelf and inserts it into the shelf. This task requires multi-view perception due to occlusions. *PutSpatulaOnTable* involves grasping a spatula on its handle from a utensil crock and placing it on the table. *PlaceAppleFromBowlIntoBin* is a long-horizon bimanual task where one arm picks up an apple from a bowl and places it on the shelf, and then the other arm picks up the apple from the shelf and places it into a bin. The simulation dataset contains 25 demonstrations per task with varying initial object poses. Each demo includes RGB-D observations from 16 camera poses, yielding 400 videos in total per task. We use 12 views for training and 4 for testing, yielding a total of 300 camera views for training and 100 camera views for testing per task. All camera poses are sampled from the upper hemisphere positioned above the workstation. More details can be found in Section A.2.

*Real-world Tasks:* The dataset contains 4 tasks (Figure 8) collected using two Franka Panda robot arms —*AddOrangeSlicesToBowl*, *PutCupOnSaucer*, *TwistCapOffBottle*, and *PutSpatulaOnTable*. Each task contains 20 robot teleoperation demonstrations. For each demonstration, we use two FRAMOS D415e cameras to capture synchronized RGB-D sequences from different viewpoints.

## 4.2 4D VIDEO GENERATION RESULTS

**Evaluation Metrics.** We evaluate the proposed method and baselines across three key aspects: RGB video generation quality, depth generation quality, and cross-view 3D point consistency.

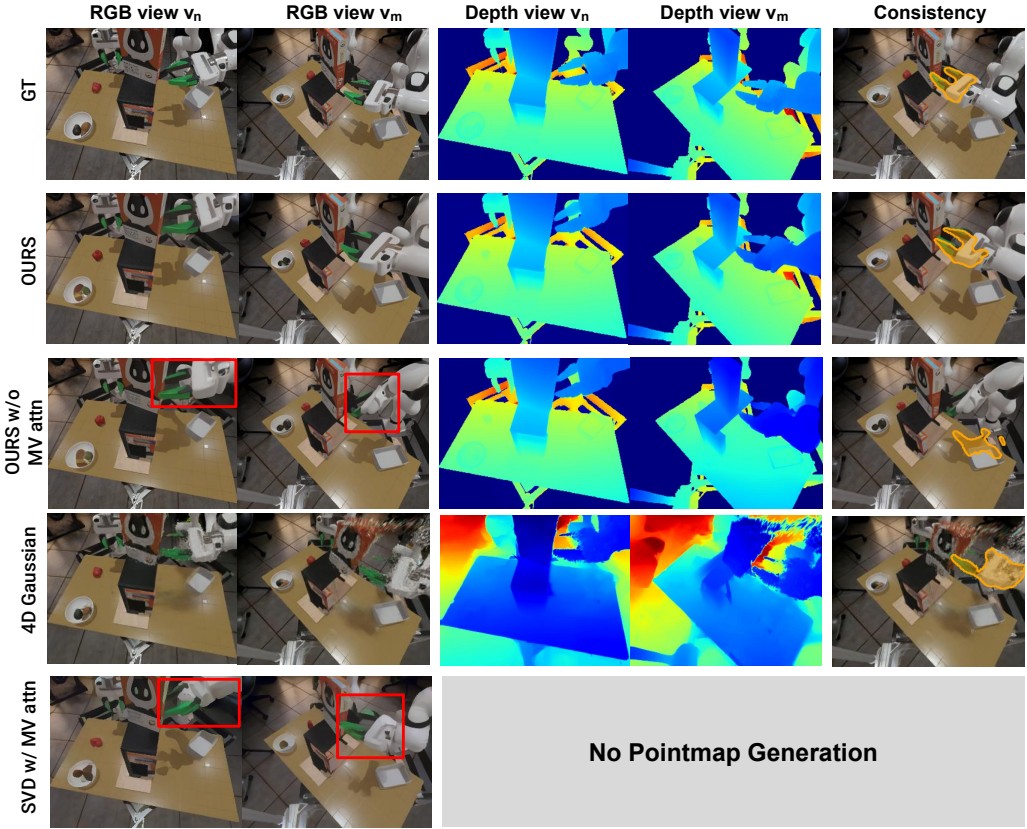

Figure 3: **Qualitative Results and Comparisons under Novel Camera Views.** Our method generates geometrically consistent 4D videos across camera views. In contrast, baseline results often exhibit significant cross-view inconsistencies or contain noticeable artifacts in the RGB or depth predictions. Video results can be found on the project website.

*Video prediction:* To evaluate RGB video generation quality, we compute the commonly used Fréchet Video Distance (Unterthiner et al., 2018) (FVD) between the generated video and ground-truth video. FVD-$n$ evaluates the prediction from the reference view $v_n$, and FVD-$m$ evaluates the predcition from view $v_m$.

*Depth prediction:* To evaluate the quality of the generated depth, we extract the z-axis values from the predicted pointmaps and compare them with the ground truth depth images. We use two standard depth evaluation metrics: absolute relative error (AbsRel $= |y - \hat{y}|/y$) and threshold accuracy ($\delta_1 = \max(\hat{y}/y, y/\hat{y}) < 1.25$), where $y$ is the ground truth depth and $\hat{y}$ is the predicted depth.

*Cross-View 3D Consistency:* To evaluate the 3D consistency of the generated pointmaps across views, we compute the mean Intersection-over-Union (mIoU) on object masks. We use SAM2 (Ravi et al., 2024) to track the robot gripper and obtain binary masks from the generated videos for both views. For each frame, we lift the gripper mask in view $v_n$ to 3D space and then re-project it to view $v_m$. We then compute the IoU between the bounding boxes of the projected and original gripper masks. The mIoU is averaged over all time steps, and higher values indicate stronger 3D alignment across views.

**Baselines.** We compare our method with prior 4D generation approaches and variants of our model to evaluate generation quality and multi-view consistency. All models are trained on the same multi-view RGB-D video dataset and **tested on novel viewpoints** sampled from 100 unseen views per task. More details can be found in Section A.2.2.

- *OURS w/o MV attn:* We remove the multi-view cross-attention mechanism in the U-Net decoder; each view is instead assigned a separate decoder with no information sharing between them.

- *4D Gaussian (Wang et al., 2024a):* A baseline method that predicts one single-view RGB video using a finetuned SVD model (Blattmann et al., 2023) on our dataset, and then use a 4D Gaussian method, Shape of Motion (Wang et al., 2024a), to reconstruct a dynamic 4D scene from the video.

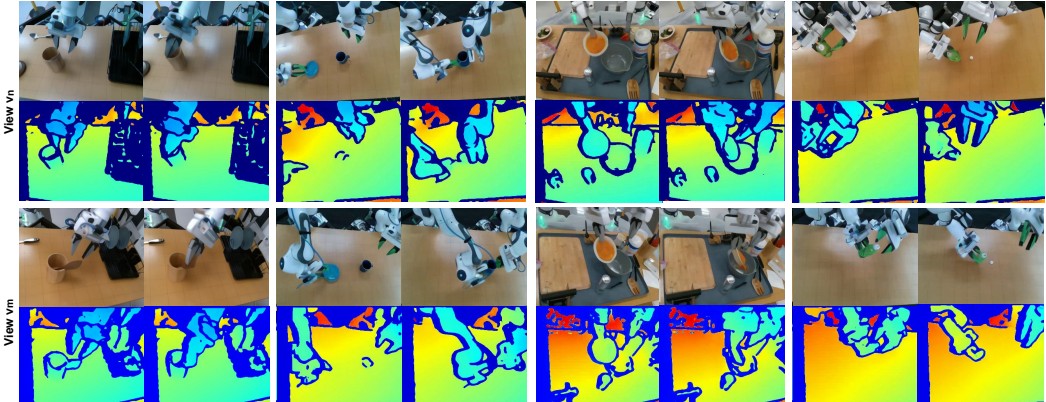

Figure 4: **Real World Multi-Task 4D Generation Results.** By fine-tuning the model trained in simulation, our model can generate high-fidelity real-world RGB-D sequences from multiple views that accurately capture the robot gripper motion over time. Video results can be found on the project website.

- *SVD (Blattmann et al., 2023):* Stable Video Diffusion is a state-of-the-art video generation model. We finetune it on our dataset to predict stereo RGB video sequences.

- *SVD w/ MV attn:* We finetune SVD on our dataset to predict stereo RGB video sequences, with additional multi-view cross-attention layers added in the U-Net decoder; similar to our method, each view is assigned a separate decoder.

**Results on Simulation Tasks.** Results are reported in Table 1, and Figure 3 compares the predictions generated by different methods for the *StoreCerealBoxUnderShelf* task. Qualitative results and comparisons for other tasks can be found in Section A.4. Our method mostly achieves the best results across all tasks. For multi-view RGB video generation, it produces lower FVD scores than all baselines in most cases, indicating high temporal coherence and visual fidelity. Additionally, our method achieves the best depth prediction and cross-view consistency scores, demonstrating that the geometry-consistent supervision effectively guides the model to generate spatially aligned and geometrically coherent videos.

We show that multi-view cross-attention is a crucial design choice for helping the model learn 3D geometric correspondences across camera views. The variant without multi-view cross-attention *OURS w/o MV attn* exhibits significantly lower 3D consistency, as measured by the mIoU metric. The video and depth generation quality also degrades in both views. In particular, the model performs poorly on the AbsRel-$m$ and $\delta_1$-$m$ metrics, indicating that it fails to learn the transformation needed to generate pointmaps in view $v_n$'s camera coordinate frame from view $v_m$ without the support of cross-attention layers. As shown in Figure 3, the gripper pose of *OURS w/o MV attn* is inconsistent across the two RGB views, and the projected gripper mask significantly misaligns with the actual gripper mask, as shown in the last column.

*4D Gaussian* performs worse in video and depth generation quality, as well as cross-view consistency, as shown in Table 1. In Figure 3, the generated RGB frames appear blurry from unseen viewpoints, and the predicted depth is inaccurate. This is because the model is optimized on one single generated RGB video with fixed camera view, making it difficult to synthesize novel views.

For the other baselines, *SVD* and *SVD w/ MV attn*, depth is not predicted and no geometric supervision is applied. While multi-view cross-attention is added to *SVD w/ MV attn* to enable information transfer between camera views, it operates over RGB features rather than pointmaps (which naturally encode 3D structure). As a result, the generated videos lack 3D geometric consistency across views. As shown in Figure 3, *SVD w/ MV attn* produces a noticeable gripper mismatch between the two views and the RGB generation quality is significantly worse than our method.

**Results on Real-world Tasks.** By fine-tuning the simulation-trained model on a multi-task real-world dataset for around 15k steps, our method is able to generate high-fidelity multi-view RGB-D sequences for several real world manipulation tasks. As shown in Table 1 and Figure 4, the generation results accurately capture both visual appearance and depth over time and consistently outperform baselines.

## 4.3 ROBOT POLICY RESULTS

We evaluate the robot policy's success rate and generalization to novel camera views across three simulation tasks. Each task is tested in 30 rollouts with unseen initial object poses and novel camera viewpoints sampled from the test dataset described in Section 4.1. The success rate of task completion is reported in Table 2.

| Method | Task 1 | Task 2 | Task 3 | Avg |
|---|---|---|---|---|
| Dreamitate (Liang et al., 2024) | 0.10 | 0.17 | 0.10 | 0.12 |
| DP (Chi et al., 2023) | 0.10 | 0.27 | 0.00 | 0.12 |
| DP3 (Ze et al., 2024) | 0.23 | 0.27 | 0.00 | 0.25 |
| OURS | **0.73** | **0.67** | **0.53** | **0.64** |

Table 2: Task Success Rate for Manipulation Tasks.

The video generation model takes RGB-D observations from two novel camera views as input and predicts future observations. The generated 4D video is then passed to the pose tracking model to extract 6DoF gripper poses for both robot arms. Gripper openness is inferred using the method described in Section 3.4. The robot executes actions in an open-loop manner: after each execution, the next inference is performed using the updated RGB-D observations. Each inference takes approximately 30 seconds to generate 10 future frames on 1 NVIDIA GeForce RTX 4090 GPU.

**Baselines.** We train baselines on the same dataset described in Section 4.1 as our method, and test on novel camera views during deployment.

_Dreamitate (Liang et al., 2024)_: a state-of-the-art video generation method for visuomotor policy. Dreamitate uses a pretrained SVD model and finetunes it on robotic task videos to generate stereo RGB video predictions. Since it does not predict depth, it employs MegaPose (Labbé et al., 2022) to extract the 6DoF pose of end effectors from the generated videos.

_Diffusion Policy (DP) (Chi et al., 2023)_: a UNet diffusion model that predicts future robot end-effector trajectories, conditioned on history RGB image observations from two camera views. We randomly sample two camera views in the training dataset and encode their corresponding RGB observations using a CLIP-pretrained ViT model (Radford et al., 2021), following the same encoder setup as in (Chi et al., 2023). The diffusion model outputs robot end effector trajectories in the next 16 steps of which the first 6 actions are executed. The model is evaluated on two unseen camera views sampled from the test dataset.

_DP3 (Ze et al., 2024)_: To compare our method with a behavior cloning method that takes in RGB-D information, we implement DP3, a UNet-based diffusion model that predicts future robot end-effector trajectories conditioned on global 3D point clouds with colors. Following the DP baseline setup, we randomly sample RGB-D images from two training camera views and aggregate their projected global point clouds as input condition. The evaluation setting is same as the DP baseline.

As shown in Table 2, _Dreamitate_ consistently underperforms across all tasks. The lack of depth prediction and geometric consistency supervision results in lower-quality, view-inconsistent video outputs, which degrade the accuracy of the extracted poses. Consequently, the downstream robot policy suffers from a high failure rate. In contrast, our method—which jointly predicts RGB-D sequences and enforces 3D consistency—achieves over 50% higher task success rate on average.

In addition, _Diffusion Policy_ struggles to generalize to unseen viewpoints, even though it is trained on demonstrations from multiple views. This is because the model does not explicitly model geometric correspondences across views and it's challenging for the model to learn view-invariant actions by simply conditioning on features extracted from multi-view images. Using global point clouds as input, _DP3_ achieves slightly better performance on the _StoreCerealBoxUnderShelf_ task, improving cereal box grasp accuracy with the added depth information. However, no gains are observed on the other two tasks, likely because the objects being manipulated are too small and the benefit of end-to-end learning from point clouds is not obvious. Most failures remain grasping failures on the spatula or apple. On the other hand, direct gripper pose tracking on generated RGB-D videos yields higher action accuracy and improves grasping success.

## 5 LIMITATION

First, our method requires RGB-D video dataset with varying camera viewpoints for training. While such datasets are easy to generate in simulation, collecting them in the real world is challenging due

to hardware constraints and camera calibration requirements. Additionally, obtaining high-quality depth in real-world settings is often difficult. Recent advances in depth estimation from RGB images (Wen et al., 2025; Wang et al., 2025) show promising results and can be leveraged to support high quality data curation in real-world environments in future work. Second, the inference speed of the current video generation model is relatively slow compared to an end-to-end behavior cloning policy, making closed-loop planning difficult. Recent flow matching (Davtyan et al., 2023; Jin et al., 2024) or autoregressive transformers (Yin et al., 2024; Deng et al., 2024; Li et al., 2024b; Gu et al., 2025) have demonstrated faster video generation speed, which could lead to more reactive robot policies.

## 6 CONCLUSION

We present a 4D video generation model that produces spatio-temporally consistent RGB-D sequences. Our method introduces geometric-consistent supervision during training by projecting pointmaps from one camera view into another to enforce cross-view consistency. By learning a shared geometric space, the model can generate future RGB-D videos from novel viewpoints without requiring camera poses at inference time. We demonstrate improved video generation quality and 3D consistency compared to baseline methods. Additionally, the generated 4D videos can be directly used to extract robot actions using an off-the-shelf 6DoF pose tracking model, enabling higher success rate on several robot manipulation tasks.

## 7 ETHICS STATEMENT

This work focuses on developing models for multi-view video generation and their application to robot manipulation. The primary goal is to advance general-purpose robotic capabilities in household and assistive settings. The datasets used in this work include synthetic data generated in simulation and real-world robot demonstrations collected in controlled lab environments; no human subjects or sensitive personal data are involved. The proposed method does not generate or rely on human biometric information. Potential societal benefits include enabling robots to better assist with daily tasks, elder care, and other beneficial applications. At the same time, we acknowledge potential misuse risks, such as manipulative content generation. To mitigate these risks, we limit the scope of this work to robotic data and will release code and models under a license that restricts harmful use.

## 8 REPRODUCIBILITY STATEMENT

We have made extensive efforts to ensure the reproducibility of our work. Details of the model architecture, training objectives, and hyperparameters are provided in Section 3 in the main paper and Section A.1, Section A.3, Section A.4 in Appendix. The datasets used in simulation and real-world experiments, including task design and camera sampling procedure, are described in Appendix Section A.2. Evaluation metrics and baseline implementations are specified in Section 4 in the main paper. We also provide a project website (`https://robot4dgen.github.io/`) with qualitative results and visualizations, and we will release code, datasets, and pre-trained models upon acceptance to further support reproducibility.

## ACKNOWLEDGMENTS

The authors would love to thank Basile Van Hoorick and Kyle Sargent for sharing their codebase for SVD fine-tuning; Ruoshi Liu and Yinghao Xu for brainstorming and technical discussions. In addition, we would like to thank all REAL lab members: Yifan Hou, Hojung Choi, Mengda Xu, Huy Ha, Mandi Zhao, Xiaomeng Xu, Austin Patel, Yihuai Gao, Chuer Pan, Haochen Shi, Haoyu Xiong, Maximilian Du, Zhanyi Sun, et al. for fruitful research discussions, feedback on paper writing, and emotional support. We would like to thank interns and collaborators at Toyota Research Institute: Alan Zhao, Chuning Zhu, Zhutian Yang, Naveen Kuppuswamy, Patrick Tree Miller, Blake Wulfe, and more, for project discussions and technical support. This work was supported in part by the Toyota Research Institute, NSF Award #2143601, #2037101, and #2132519, Sloan Fellowship. The views and conclusions contained herein are those of the authors and should not be interpreted as necessarily representing the official policies, either expressed or implied, of the sponsors.

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

## A    APPENDIX

In Section A.1, we provide more details of our 4D generation model architecture. In Section A.2, we describe the camera sampling method used to generate the multi-view RGBD video dataset for training the 4D generation model. In Section A.3, we provide details such as compute resources requirements and hyperparameter choices for model training and inference. In Section A.4, we provide more quantitative and qualitative results of our method and additional baselines. In Section A.5, we discuss the LLM usage of our work.

### A.1    MODEL DETAILS

In §3.1 and §3.2, we discussed that the model takes in historical video frames $\{\mathbf{O}_{t-h+1}, \cdots, \mathbf{O}_t\}$ and historical pointmaps $\{\mathbf{X}_{t-h+1}, \cdots, \mathbf{X}_t\}$ from both the native view $v_n$ and the second view $v_m$. In practice, we use the latest observation, and repeat it $h = 10$ times to match the number of frames that needs to be predicted, following the implementation in SVD (Blattmann et al., 2023).

Each pair of RGB image and pointmap condition $\mathbf{O}_t^v, \mathbf{X}_t^v$, where $v \in \{n, m\}$, is independently encoded using separate VAE encoders for images and pointmaps, as detailed in §3. The image VAE encodes each RGB frame into a latent feature of shape $h \times c \times w' \times h'$, where $h = 10$ is the temporal horizon, $c=4$ is the latent channel size, $w'=32$ and $h'=40$ are spatial dimensions of the latent feature maps. Similarly, the pointmap VAE encodes pointmap into shape $h \times c \times w' \times h'$. These encoded image and pointmap features are then concatenated along the channel axis with the corresponding noisy latents of future images and pointmaps, yielding a combined input tensor of shape $h \times 4c \times w' \times h'$ ($h \times 16 \times 32 \times 40$), which is fed into the U-Net diffusion model.

To allow information sharing between the two diffusion branches as shown in Figure 2, we add one cross-attention layer after each decoder block in the U-Net diffusion model for the branch corresponding to view $v_m$. This results in 12 added cross attention layers. As illustrated in Figure 5, the query to the cross-attention layer are feature map tokens (feature at each pixel in the feature map) output by the decoder block in view $v_m$, where $c'$ is the feature dimension, $h'$ and $w'$ are spatial dimensions of the feature map; the key and value are feature map tokens output by the corresponding decoder block in the native view $v_n$'s branch. The updated features are passed to the next decoder block in view $v_m$. The cross-attention layers capture spatial correspondences between the views through our geometric-consistent supervision mechanism.

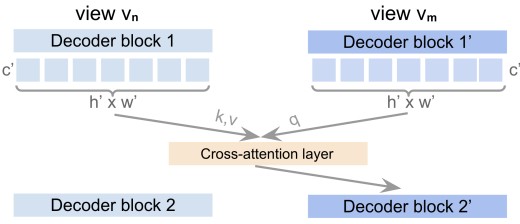

Figure 5: **Multi-View Cross-Attention.** We insert a cross attention layer after each decoder block in the U-Net diffusion model for view $v_m$. By cross-attending to features in the native view $v_n$, the cross-attention layers allow information sharing between view branches.

We use pre-trained weights in SVD to initialize the denoising U-Net model and find that using pre-trained weights helps the model converge faster. To get better prediction quality around the robot gripper, which is important for action extraction later, we apply a re-weighting mechanism in the diffusion loss. Concretely, we use binary masks (in simulation, the object segmentations are provided; in real world, we use SAM2 (Ravi et al., 2024)) of the robot gripper region and downsample it by a factor of 8 to match the resolution of the latent space while still keeping the spatial correspondence. The resulting downsampled masks provide a spatial weight map at each timestep $t'$, denoted as $w_g(t')$, which is incorporated into the joint diffusion loss mentioned in §3.3 as follows:

$$\mathcal{L} = \sum_{t'=t+1}^{t+h} \left[ \left(1 + \mathbb{1}_{\{w_g(t')=1\}}\right) \cdot \left( \underbrace{\mathcal{L}_{\text{diff}}^n(t') + \mathcal{L}_{\text{diff}}^m(t')}_{\text{RGB loss}} + \lambda \cdot \underbrace{\mathcal{L}_{\text{3D-diff}}(t')}_{\text{pointmap loss}} \right) \right] \tag{4}$$

where $\mathbb{1}_{\{w_g(t')=1\}}$ is an indicator function that activates if the pixel value on the spatial weight map is 1 and 0 otherwise. We add this indicator to a base weight of 1, effectively doubling the contribution of

loss terms at gripper regions. This weighting encourages higher prediction accuracy in areas critical for gripper pose estimation in the policy extraction phase.

## A.2 DATASET DETAILS

### A.2.1 SIMULATION TASKS

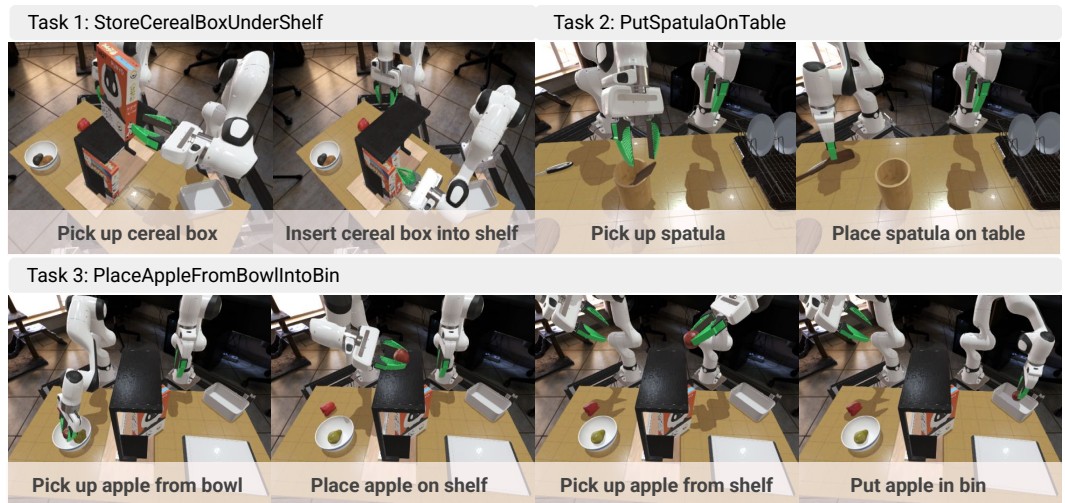

Figure 6: **Simulation Tasks for Evaluation.**

### A.2.2 CAMERA SAMPLING

To sample cameras for rendering multi-view RGBD videos, we first sample camera positions within a half-sphere shell defined by an inner radius ($r_1 = 0.7m$) and outer radius ($r_2 = 1.2m$), with the center being the origin of the world coordinate system (center of the table). We restrict the range of the camera positions within the area between $0.2m \leq x \leq 0.6m$, $-0.5m \leq y \leq 0.5m$, and $0.7 \leq z \leq 1.2m$, as shown in Figure 7 (a-c). The world coordinate system is shown in Figure 7 (d).

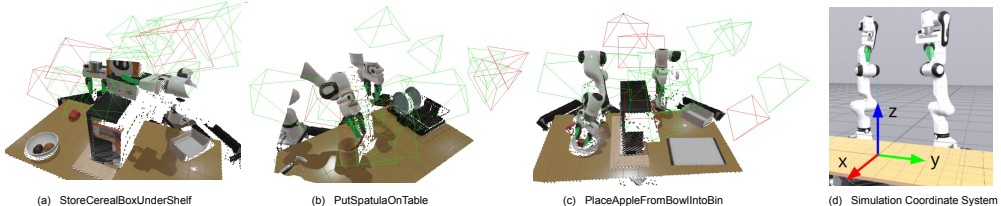

(a) StoreCerealBoxUnderShelf  (b) PutSpatulaOnTable  (c) PlaceAppleFromBowlIntoBin  (d) Simulation Coordinate System

Figure 7: **Camera Sampling Visualization.** We randomly sample 16 camera poses per episode using our proposed technique. (a)-(c) show example camera poses for each task, with green cameras used for training and red for evaluation. (d) shows the simulation world coordinate frame.

## A.2.3 REAL WORLD TASKS

Figure 8: **Tasks in Real-World Dataset.**

## A.3 TRAINING DETAILS

The 4D generation model described in § 3 is trained separately for each task in § 4 for approximately 60 epochs using 4 NVIDIA RTX A6000 GPUs (48GB memory each). We fine-tune the full U-Net backbone of the SVD (Blattmann et al., 2023) model with a learning rate of $1 \times 10^{-6}$, using the AdamW optimizer ($\beta_1 = 0.95$, $\beta_2 = 0.999$, $\epsilon = 10^{-8}$, weight decay $= 1 \times 10^{-6}$) and a batch size of 4. The image and pointmap VAE encoders are frozen during diffusion model training.

At inference time, we apply the standard EulerEDMSampler (Karras et al., 2022) with 25 denoising steps. For robot policy deployment, both the generation model and the pose tracking model are run on a single NVIDIA GeForce RTX 4090 GPU.

| Method | Inference Time | Training Memory | # of Trainable Parameters |
|---|---|---|---|
| OURS | 30.0 s | 47 G | 2.4 B |
| OURS w/o MV attn | 29.3 s | 46.5 G | 2.38 B |
| 4D Gaussian | 2 s | 2813 M | 856,774 |
| SVD (Dreamitate) | 13.4 s | 45.8 G | 1.54 B |
| SVD w/ MV attn | 15.1 s | 46.3 G | 2.4 B |

Table 3: Quantitative comparisons with baselines in terms of inference time, GPU memory / compute resource consumption and parameter count.

We report quantitative comparisons with baselines in terms of inference latency, GPU memory / compute resource consumption, and parameter count. Our method requires 30.0s inference time with 47G training memory and 2.4B parameters: adding multi-view cross attention layers only slightly increases each metric yet yields more geometry-consistent and high-quality generation results. In contrast, 4D Gaussian is significantly faster (2s) and lighter (2813M memory, 856,774 parameters), but produces poor quality results for novel view rendering and also requires a generated single-view video as input, while our method directly predicts future frames based on a single RGB-D image per view.

We acknowledge that the higher inference time reflects the cost of predicting both RGB and depth modalities, but this trade-off results in stronger video fidelity and geometric consistency. Looking ahead, improvements in efficiency can be explored by applying the geometry consistency objective to faster generative models, such as leveraging 3D VAEs (Zhao et al., 2024) to reduce temporal dimensions of the letent tokens, adopting one-step or few-step diffusion models (Lin et al., 2025), or employing autoregressive transformers with lower latency (Huang et al., 2025b). On the policy side, a hierarchical structure (Black et al., 2023; Wen et al., 2023) could further mitigate the cost of dense video inference: instead of extracting and executing all poses from generated RGB-D frames, the model can act as a high-level planner where sparse predicted frames serve as subgoals, while a low-level controller or learned policy fills in reactive actions. This design would reduce the number of generated frames required for action planning, thereby accelerating inference while still preserving the advantages of geometry-aware video generation for downstream robotic tasks.

## A.4 ADDITIONAL RESULTS

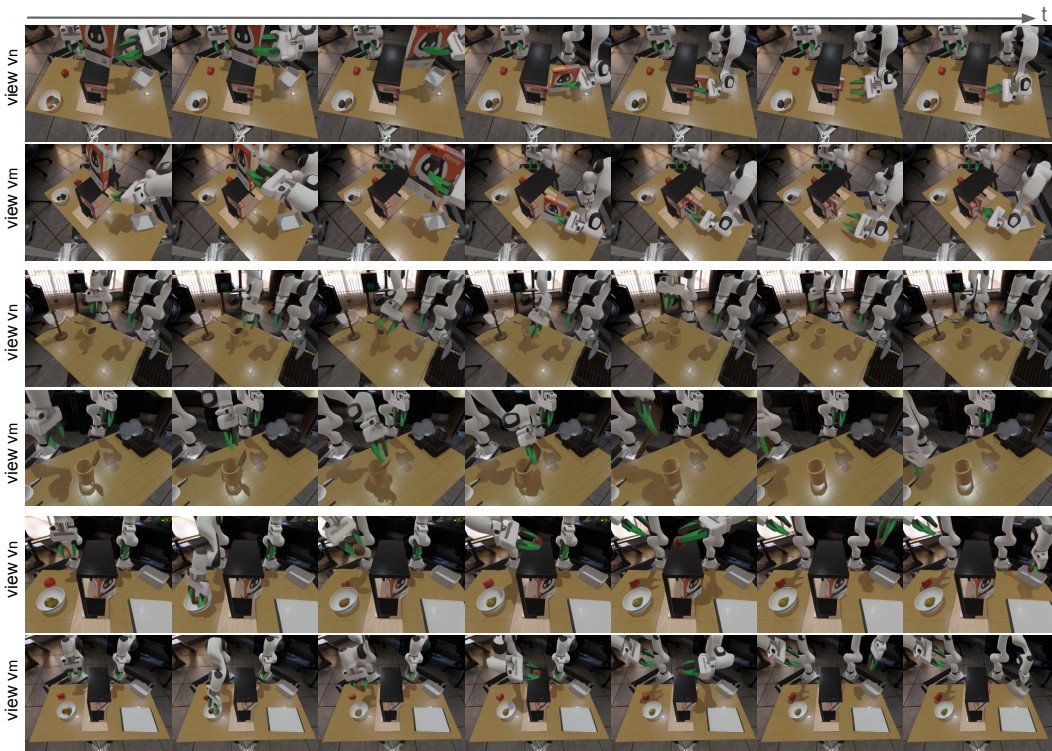

Figure 9: **Qualitative Multi-View Video Generation Results in Simulation.** We show temporal results generated by our 4D video generation model across three robot manipulation tasks. With geometry-consistent supervision and joint temporal and 3D consistency optimization, our model is able to output spatio-temporally consistent videos across camera views with high visual fidelity in unseen views.

In Figure 9 and Figure 10, we show generated RGB video sequences on both simulation and real world tasks using our proposed 4D generation model. With geometry-consistent supervision and joint temporal and 3D consistency optimization, our model is able to output spatio-temporally consistent videos across camera views with high visual fidelity. We also show baseline comparison results on the *PlaceAppleFromBowlIntoBin* task in Figure 11 and *PutSpatulaOnTable* task in Figure 12. Our method consistently achieves the best RGB video and depth generation quality, with high multi-view consistency. Baseline results often exhibit significant cross-view inconsistencies (marked in red) or contain noticeable artifacts in the RGB or depth predictions.

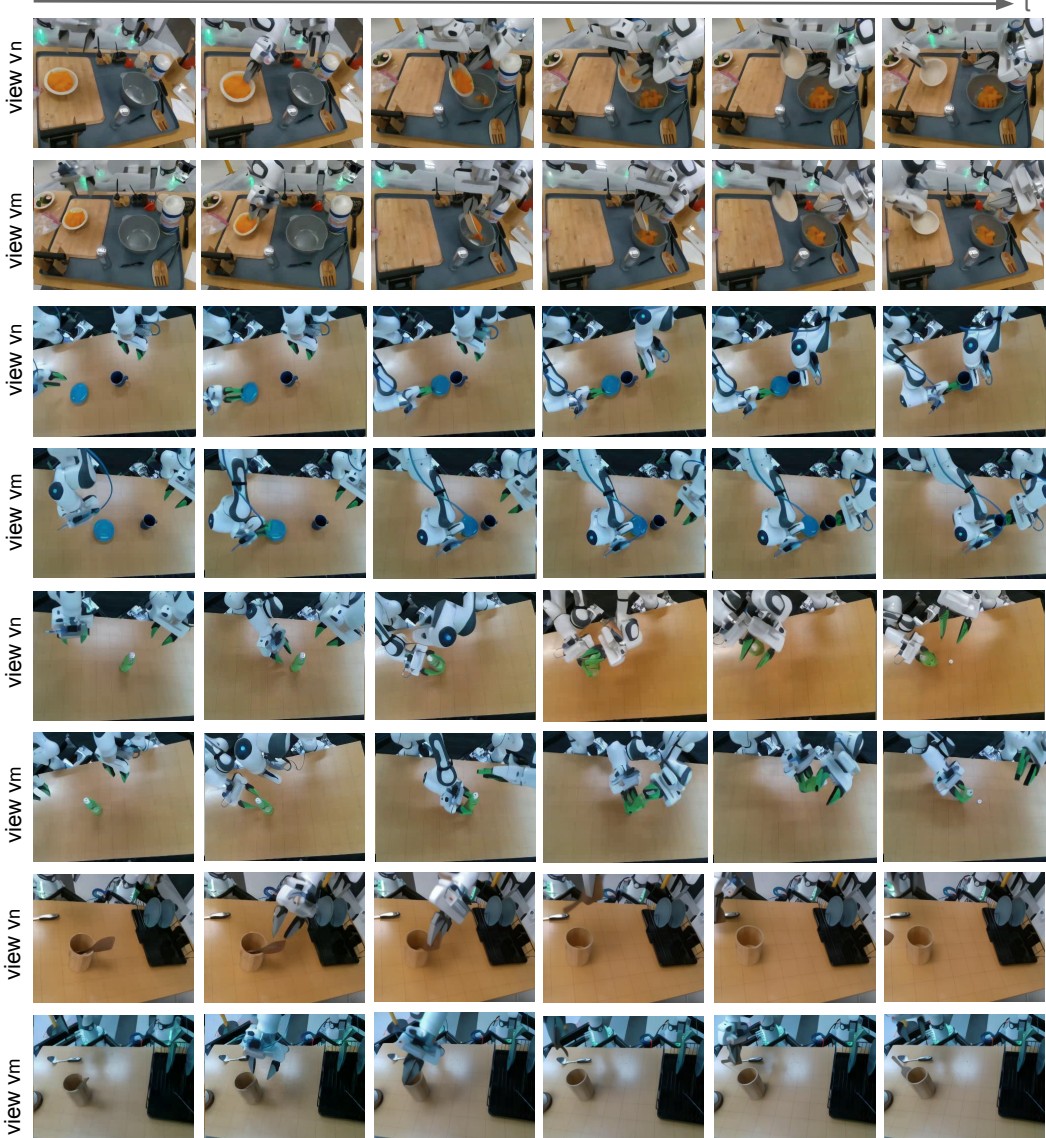

Figure 10: **Qualitative Multi-View Video Generation Results in Real World.** We show temporal results generated by our 4D video generation model across four real-world robot manipulation tasks. By fine-tuning the simulation-trained model for around 15k steps, our model is able to output spatio-temporally consistent videos across camera views with high visual fidelity.

## A.5 THE USE OF LARGE LANGUAGE MODELS (LLMs)

We used large language models (LLMs), specifically OpenAI's ChatGPT 5, to assist with writing and editing the manuscript. The models were used primarily for improving the clarity, grammar, and readability of the text, including rephrasing sentences, polishing explanations, and standardizing style. Anthropic's Claude was used to assist in generating visualization-related code, such as for camera pose and depthmap visualization in the figures. All technical content, experimental design, analysis, and conclusions were conceived, implemented, and validated by the authors without the use of LLMs. The authors have carefully reviewed and verified all content produced with the assistance of LLMs to ensure accuracy and originality.

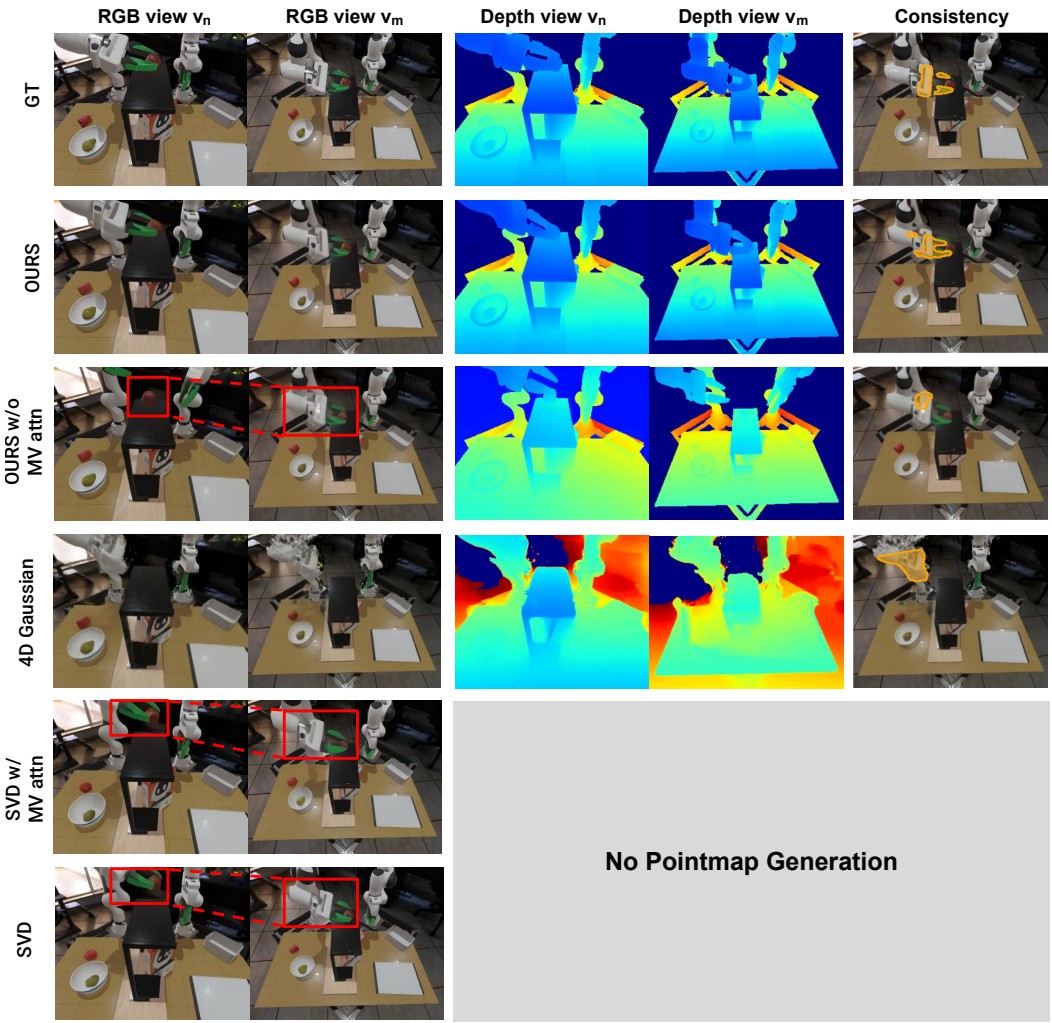

Figure 11: **Qualitative Results of PlaceAppleFromBowlIntoBin task.** Our method achieves the best RGB video and depth generation quality, with high multi-view consistency. Baseline results often exhibit significant cross-view inconsistencies (marked in red) or contain noticeable artifacts in the RGB or depth predictions.

## A.6 POSE TRACKING ERROR ANALYSIS

The pose tracking error arises from two sources: (1) the RGB-D generation quality of the video model, and (2) the detection accuracy of FoundationPose. To disentangle these factors, we additionally evaluate FoundationPose directly on ground-truth RGB-D images to isolate its performance independently of any generation errors. As shown in Figure 13, we visualize one representative trajectory from each task and report the per-action MSE between the detected pose and the ground-truth pose throughout the task. We also include the frames with the top five highest pose-tracking errors to illustrate the characteristic failure cases.

We observe that the MSE error of pose tracking is generally around or below 0.001 for both grippers across tasks, indicating good pose-tracking accuracy. The only notable exception is the right gripper in the PutSpatulaOnTable task and the left gripper in StoreCerealBoxUnderShelf towards the end of the task, where the error occasionally peaks at around 0.003.

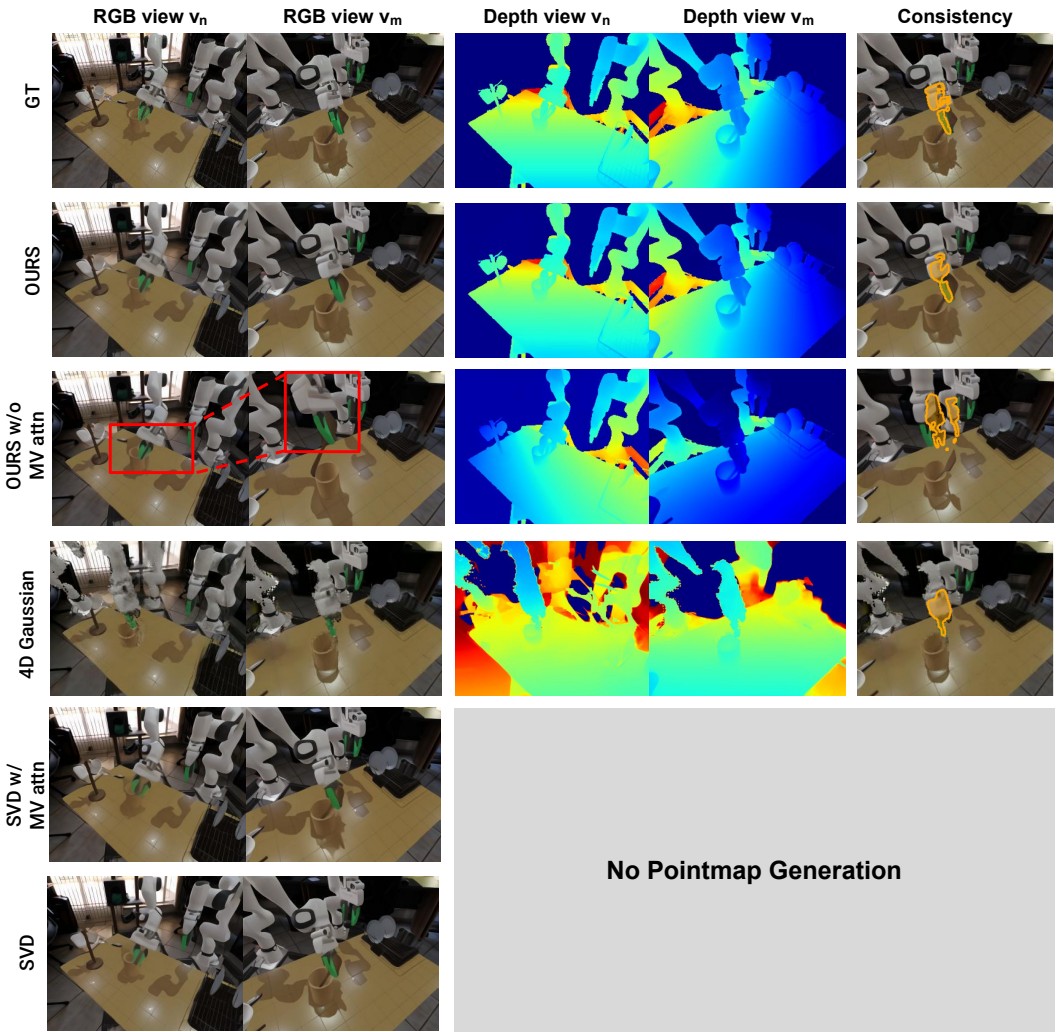

Figure 12: **Qualitative Results of PutSpatulaOnTable task.** Our method achieves the best RGB video and depth generation quality, with high multi-view consistency. Baseline results often exhibit significant cross-view inconsistencies (marked in red) or contain noticeable artifacts in the RGB or depth predictions.

## A.7 SCALING TO MORE INPUT VIEWS

Our main evaluation focuses on simultaneously generating two views. We additionally study how the proposed framework scales when incorporating more input views at inference time.

Without retraining, the model can incorporate a third (or more) view(s) at inference time by performing an additional forward pass between the reference view and each new view. This approach increases inference time approximately linearly with the number of views, but does not require architectural modifications or additional training. Designing more efficient mechanisms for large-scale multi-view inference is an interesting direction for future work, similar in spirit to how Fast3R (Yang et al., 2025a) extends DUSt3R to handle thousands of input images.

| View | mIoU | FVD | AbsRel | $\delta_1$ |
|---|---|---|---|---|
| View 1 | – | 515.53 | 0.06 | 0.95 |
| View 2 (w/ View 1) | 0.62 | 566.86 | 0.09 | 0.95 |
| View 3 (w/ View 1) | 0.54 | 662.83 | 0.12 | 0.89 |

Table 4: Performance generating three views simultaneously on *StoreCerealBoxUnderShelf*.

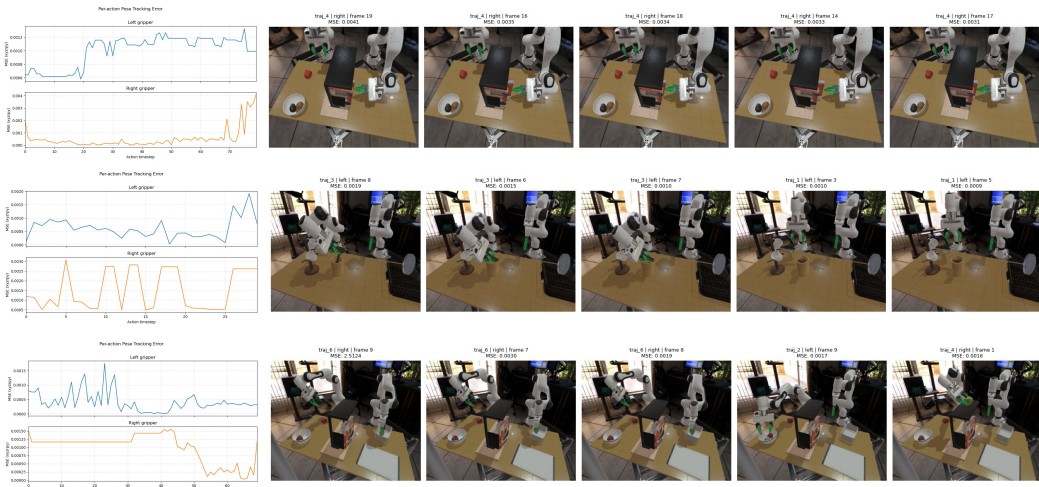

Figure 13: **Analysis of 6-DoF pose-tracking errors.** Left: per-action MSE curves for left and right grippers across one representative trajectory from each of the three tasks. Right: visual examples of the five highest-error frames for each task.

**Three-view Evaluation.** We evaluated the performance of incorporating a third predicted view on the StoreCerealBoxUnderShelf task. In this setting, the model simultaneously generates videos in three randomly sampled, unseen viewpoints, and the reported metrics are averaged across different initial RGB-D conditions.

Table 4 summarizes the results. Although performance degrades slightly, the model is still able to produce RGB-D videos with good visual quality and cross-view consistency.

