# OpenReview forum: "Geometry-aware 4D Video Generation for Robot Manipulation"
_ICLR.cc/2026/Conference — ICLR 2026 Poster_

### Official Review · Reviewer_RXK1 · 2025-10-25

**Soundness:** 2
**Presentation:** 3
**Contribution:** 2
**Rating:** 4
**Confidence:** 3

**Summary:**

This paper proposes a geometry-aware 4D video generation model tailored for robot manipulation, which enforces multi-view 3D consistency during training via cross-view pointmap alignment and leverages pretrained video diffusion models for temporal coherence, enabling the generation of spatio-temporally aligned RGB-D sequences from novel viewpoints without camera pose inputs. It evaluates the model on both simulated (e.g., StoreCerealBoxUnderShelf) and real-world (e.g., TwistCapOffBottle) robotic tasks. Additionally, the predicted 4D videos can be used with an off-the-shelf 6DoF pose tracker (e.g., FoundationPose) to extract robot end-effector trajectories.

**Strengths:**

1. Enables geometry-consistent 4D video generation for multi-object dynamic robot manipulation scenes (a gap in prior 3D-aware methods limited to single objects and static backgrounds) by enforcing cross-view pointmap alignment during training, ensuring spatio-temporal consistency across novel camera viewpoints without relying on camera poses at inference.
2. Integrates pretrained video diffusion models’ temporal priors with a dedicated geometry-consistent loss for pointmaps, achieving joint optimization of RGB video quality, depth accuracy, and multi-view 3D consistency.
3. Bridges 4D video generation to practical robot manipulation: predicted RGB-D sequences can extract robot end-effector trajectories via an off-the-shelf 6DoF pose tracker (e.g., FoundationPose) and infer gripper open/close states, leading to higher task success rates than visuomotor policy baselines.

**Weaknesses:**

1. Relies on multi-view RGB-D datasets with varied camera viewpoints for training, which are challenging to collect in real-world settings due to hardware constraints and calibration requirements, and high-quality depth data acquisition in real scenarios remains difficult.
2. The inference speed of the video generation model is relatively slow, making closed-loop planning for robot manipulation impractical compared to end-to-end behavior cloning policies.
3. The baseline comparisons lack direct competition with state-of-the-art methods specifically designed for multi-object dynamic robot manipulation scenarios; selected baselines (e.g., 4D Gaussian, SVD) either have different task scopes or lack 3D consistency modeling, weakening the persuasiveness of performance superiority.
4. Key implementation details for reproducibility are insufficiently disclosed, such as the specific camera sampling parameters (e.g., exact pitch/yaw ranges), training hyperparameter schedules (e.g., learning rate decay strategy), and threshold values for gripper state inference (e.g., distance threshold δ), which may hinder result validation.
5. The multi-view cross-attention mechanism, a core component for 3D consistency, lacks unique design details; it is not clarified how it adapts to pointmap geometric features (e.g., whether attention weights correlate with 3D distances), making it indistinguishable from generic cross-attention modules.

**Questions:**

see weaknesses

---

> ### Author Response · Authors · 2025-11-19
> **Response to reviewer RXK1**
>
> We would like to thank the reviewer for the constructive feedback. Here are the responses to the weaknesses identified.
>
> > Relies on multi-view RGB-D datasets with varied camera viewpoints for training, which are challenging to collect in real-world settings due to hardware constraints and calibration requirements, and high-quality depth data acquisition in real scenarios remains difficult.
>
> We agree with the reviewer on this point. Though we foresee in future the data requirement can be easier to fulfill through: 1) photorealistic simulation environments such as Behavior 1K [1] with less sim2real gap; 2) high quality metric depth prediction models such as FoundationStereo [2], where we can avoid depth collection in real world by predicting metric depths from RGB images; 3) real world robot data collection platform such as DROID [3], with adjustable RGBD camera mounts. Together, these approaches can make the required datasets more accessible.
>
> > The inference speed of the video generation model is relatively slow, making closed-loop planning for robot manipulation impractical compared to end-to-end behavior cloning policies.
>
> We agree with the reviewer that this is a valid concern, and we acknowledge that the SVD-based model used in this paper is not optimized for inference speed. However, there are several ways to make video models better suited for robot manipulation tasks.
>
> One way is to make video models faster. To do so, we can leverage (a) a 3D VAE [5] to reduce the temporal dimension of the input latents to the diffusion model, (b) one-step or few-step diffusion models [6] or autoregressive transformers [7] with faster inference latency.
>
> Another way is to better integrate the outputs of the video models with the downstream policy. Instead of directly extracting poses from the dense RGB-D frames and executing them, we can adopt a hierarchical task execution structure similar to [8] and [9], where we use the video generation model as a high-level planner where the generated sparse frames serve as ‘subgoals’.  And then we can use a low-level controller or learnable policies to generate more reactive actions to fill in the gap between subgoals. In this case, we can reduce the number of video frames that need to be generated in one forward pass and make the policy more reactive to local changes.
> Another way is to directly condition the latent features predicted by the latent diffusion model to a downstream BC policy, however, this approach requires a moderate amount of robot data with dense action labeling to train the entire network end to end.
>
> > The baseline comparisons lack direct competition with state-of-the-art methods specifically designed for multi-object dynamic robot manipulation scenarios; selected baselines (e.g., 4D Gaussian, SVD) either have different task scopes or lack 3D consistency modeling, weakening the persuasiveness of performance superiority.
>
> To the best of our knowledge, at the time this work was conducted, there were no video generation models specifically designed or trained for robot manipulation scenarios. The closest is Dreamitate, which fine-tunes SVD on their self-collected robot dataset – conceptually similar to our SVD baseline (Table 1, Page 6). For policy comparison, we benchmark against Diffusion Policy and DP3 (Table 2, Page 9), two state-of-the-art behavior cloning methods widely used for multi-object manipulation tasks, and our policy achieves higher success rate in novel views using the same training datasets as the baselines.
>
> > Key implementation details for reproducibility are insufficiently disclosed, such as the specific camera sampling parameters (e.g., exact pitch/yaw ranges), training hyperparameter schedules (e.g., learning rate decay strategy), and threshold values for gripper state inference (e.g., distance threshold δ), which may hinder result validation.
>
> We specified the camera sampling strategy and parameters in Appendix A.2.2; training parameters in Appendix A.3. We added details for the gripper detection threshold in Line 267-268. Please see the updated manuscript.
> To further facilitate reproducibility, the codebase and data used in the project will be released.

---

> ### Author Response · Authors · 2025-11-19
> **Response to reviewer RXK1 (pt. 2)**
>
> > The multi-view cross-attention mechanism, a core component for 3D consistency, lacks unique design details; it is not clarified how it adapts to pointmap geometric features (e.g., whether attention weights correlate with 3D distances), making it indistinguishable from generic cross-attention modules.
>
> Our intuition is inspired by DUSt3R, where the model’s cross-attention layers do not explicitly encode 3D distances, yet the network is still able to predict unified pointmaps across views by training on datasets containing image pairs from different viewpoints. This is achieved by expressing the pointmap observed in view v_m​ in the coordinate frame of a reference view v_n. In doing so, the cross-attention mechanism implicitly learns the relative camera transformation between the two views. Empirically, we find this mechanism to be simple yet effective to enforce spatio-temporal consistency. Future work could potentially add additional 3D distance encoding similar to the approach in MVDiffusion [4], though this probably requires good correspondence mapping across views.
>
> ---
> **References:**
>
> [1] Li, Chengshu, et al. "Behavior-1k: A human-centered, embodied ai benchmark with 1,000 everyday activities and realistic simulation." arXiv preprint arXiv:2403.09227 (2024).
>
> [2] Wen, Bowen, et al. "Foundationstereo: Zero-shot stereo matching." Proceedings of the Computer Vision and Pattern Recognition Conference. 2025.
>
> [3] Khazatsky, Alexander, et al. "Droid: A large-scale in-the-wild robot manipulation dataset." arXiv preprint arXiv:2403.12945 (2024).
>
> [4] Tang, Luming, et al. "Emergent correspondence from image diffusion." Advances in Neural Information Processing Systems 36 (2023): 1363-1389.
>
> [5] Zhao, Sijie, et al. "Cv-vae: A compatible video vae for latent generative video models." Advances in Neural Information Processing Systems 37 (2024): 12847-12871.
>
> [6] Lin, Shanchuan, et al. "Diffusion adversarial post-training for one-step video generation." arXiv preprint arXiv:2501.08316 (2025).
>
> [7] Huang, Xun, et al. "Self Forcing: Bridging the Train-Test Gap in Autoregressive Video Diffusion." arXiv preprint arXiv:2506.08009 (2025).
>
> [8] Black, Kevin, et al. "Zero-shot robotic manipulation with pretrained image-editing diffusion models." arXiv preprint arXiv:2310.10639 (2023).
>
> [9] Wen, Chuan, et al. "Any-point trajectory modeling for policy learning." arXiv preprint arXiv:2401.00025 (2023).

---

### Official Review · Reviewer_b1Za · 2025-10-26

**Soundness:** 3
**Presentation:** 3
**Contribution:** 3
**Rating:** 6
**Confidence:** 3

**Summary:**

This paper proposes a geometry-aware 4D video generation model for robot manipulation. The method introduces cross-view pointmap alignment as a geometric supervision during training. The geometric constraint allows the model to learn a shared 3D scene representation, enabling the joint optimization of spatial and temporal consistency.

**Strengths:**

1. The method achieves geometric consistency in the generated videos across multiple views through cross-view geometric supervision and cross-attention mechanism.
2. The model shows a significant improvement in task success rate for manipulation tasks compared to baseline methods.
3. The generated 4D videos can be directly combined with an off-the-shelf 6DoF pose tracker to extract robot end-effector trajectories.

**Weaknesses:**

1. The authors need to fully explain the difference between their proposed method and the joint spatio-temporal consistency optimization methods mentioned in references [3, 4]. The authors should specifically show how the proposed approach is better suited for the multi-object, dynamic robot manipulation scenes.
2. In Table 1, the $FVD-{n}$ scores for Task 2 and Task 4 are not significantly different from the SVD finetuning baseline. More results from additional tasks are needed to verify the advantage of the proposed method over SVD finetuning.
3. The current experiments mainly focus on rigid object manipulation, lack experimental evaluation on non-rigid objects (e.g., cloth).

**Questions:**

1. The current method uses geometric supervision via feature passing and cross-attention from view $v_n$ to view $v_m$. Have the authors investigated the effect of $v_m \to v_n$ or $v_n \rightleftarrows v_m$ on geometric consistency and final generation quality? Additionally, a sensitivity analysis on $\lambda$ should be included to determine if there is a trade-off between FVD and mIoU performance.
2. Since the current evaluation is primarily based on two views, what is the performance of this framework when the number of input views is increased (e.g., three or four views)?

---

> ### Author Response · Authors · 2025-11-19
> **Response to reviewer b1Za**
>
> We appreciate the reviewer’s positive assessment of our method and results. Below, we respond to the identified weaknesses and questions.
>
> **Weakness:**
> > The authors need to fully explain the difference between their proposed method and the joint spatio-temporal consistency optimization methods mentioned in references [3, 4]. The authors should specifically show how the proposed approach is better suited for the multi-object, dynamic robot manipulation scenes.
>
> ***Training domain.***  While SV4D and VividZoo focus on multi-view videos of single 3D dynamic objects, our model is trained directly on RGB-D videos of robot manipulation, which include richer dynamics such as robot–object interaction and multi-object motion.
>
> ***Camera pose requirement.*** SV4D and VividZoo require camera poses as inputs during both training and inference. In contrast, our method uses camera poses only during training to convert pointmaps into a unified coordinate frame; no pose information is needed at deployment. Furthermore, SV4D operates on monocular input videos and focuses on re-rendering those videos in novel viewpoints—a different task from ours, which predicts future RGB-D frames conditioned on a single initial RGB-D observation.
>
> ***Geometry supervision mechanism.*** To ensure geometry consistency, Vividzoo introduces 2D-3D alignment layers in addition to self-attention layers across views. We take an alternative approach where we explicitly supervise the 3D pointmap predictions of different views in one unified coordinate frame. Our approach is more memory efficient as there are less layers introduced. SV4D involves an additional 4D reconstruction procedure to fuse the predicted views, with no explicit geometry consistency supervision in the video generation process, whereas OURS combines generation with reconstruction through the pointmap alignment mechanism.
> We’ve updated the introduction section (Line 60 - 66) to reflect these differences. Please see the revised manuscript.
>
> > In Table 1, the  scores for Task 2 and Task 4 are not significantly different from the SVD finetuning baseline. More results from additional tasks are needed to verify the advantage of the proposed method over SVD finetuning.
>
> For Task 4 (multi-task real world dataset), the FVD score is almost halved from the SVD finetuning baseline for both views. We agree that for Task 2 the FVD score is not significantly better than the SVD fine tuning baseline. Yet, FVD is only one aspect of evaluation. A key advantage of our approach is that it jointly predicts RGB and depth with enforced multi-view geometric consistency. This enables stable, view-aligned 4D outputs that support downstream policy extraction via pose tracking—which results in higher task success rate as shown in Table 2 compared to using RGB only video generation and pose tracking methods such as ones used in [Dreamitate]. Thus, even though the FVD improvement is relatively modest for one of the tasks, the additional geometric information and consistency offered by our model provide meaningful benefits for manipulation.
>
> **Questions:**
> > The current method uses geometric supervision via feature passing and cross-attention from view  to view . Have the authors investigated the effect of $v_m$ -> $v_n$ or $v_m$ <-> $v_n$ on geometric consistency and final generation quality? Additionally, a sensitivity analysis on $\lambda$ should be included to determine if there is a trade-off between FVD and mIoU performance.
>
> In our experiments, adding bi-directional cross-attention degraded performance, likely due to the increased learning complexity it introduces. We find that a uni-directional design is sufficient for learning the relative pointmap transformation across views. Noteably, the direction $v_n$ -> $v_m​$ is chosen so that the reference view $v_n$ provides contextual information in the reference frame; reversing this direction is less meaningful because we do not want the second view to affect or overwrite the features of the reference view as the final pointmaps are represented in the reference view’s coordinate frame.
>
> We agree that analyzing the effect of $\lambda$ on both RGB and pointmap generation is important. We are currently running a sweep over $\lambda \in \\{0, 0.25, 0.5, 0.75, 1\\}$. Note that $\lambda = 1$ is our current version, and $\lambda = 0$ is equivalent to the [SVD] baseline. Each checkpoint requires roughly 2-3 days of training, so the full set of results will take a few more days to complete. We will update the numbers here as they become available and include the full analysis in Appendix.

---

> > ### Author Response · Authors · 2025-11-19
> > **Response to reviewer b1Za (pt. 2)**
> >
> > > Since the current evaluation is primarily based on two views, what is the performance of this framework when the number of input views is increased (e.g., three or four views)?
> >
> > Although our evaluation focuses on simultaneously generating two views, it is also possible to incorporate a third view (or more) at inference time by running an additional forward pass on the reference view and the third view, without retraining the model—though this would roughly double the inference time. Future work may explore more efficient scaling to multiple views, similar to how Fast3R [1] extends DUSt3R to handle over a thousand input images compared to DUSt3R’s stereo setting.
> >
> > We evaluated the performance of incorporating a third predicted view on the StoreCerealBoxUnderShelf task. In this setting, the model simultaneously generates videos in three randomly sampled, unseen viewpoints, and the reported metrics are averaged across different initial RGB-D conditions. Although performance degrades slightly, the model is still able to produce RGB-D videos with good visual quality and cross-view consistency.
> >
> > | View                  | mIoU | FVD    | AbsRel | $\delta_1$   |
> > |-----------------------|------|--------|--------|------|
> > | View 1                | -    | 515.53 | 0.06   | 0.95 |
> > | View 2 (w/ view 1)    | 0.62 | 566.86 | 0.09   | 0.95 |
> > | View 3 (w/ view 1)    | 0.54 | 662.83 | 0.12   | 0.89 |
> >
> > ---
> > **References:**
> >
> > [1] Yang, Jianing, et al. "Fast3r: Towards 3d reconstruction of 1000+ images in one forward pass." Proceedings of the Computer Vision and Pattern Recognition Conference. 2025.

---

### Official Review · Reviewer_ctzc · 2025-11-01

**Soundness:** 4
**Presentation:** 4
**Contribution:** 4
**Rating:** 8
**Confidence:** 4

**Summary:**

This paper proposes a novel 4D video generation framework that jointly enforces temporal coherence and multi-view 3D geometric consistency for robotic manipulation tasks. The core idea is to supervise a diffusion-based video generation model using cross-view pointmap alignment, inspired by DUSt3R, to ensure that predicted RGB-D sequences from different camera views correspond to a shared 3D scene representation. The method does not require camera poses at inference time, yet it generalizes to novel viewpoints. The authors demonstrate that the generated 4D videos can be used with off-the-shelf 6DoF pose trackers (e.g., FoundationPose) to extract accurate robot end-effector trajectories, enabling successful execution of manipulation policies in both simulation and real-world settings. Experiments across three simulated and four real-world tasks show consistent improvements over strong baselines in video quality, depth accuracy, and cross-view consistency.

**Strengths:**

1. Originality: The integration of cross-view pointmap alignment into a video diffusion model for 4D generation is novel. Unlike prior 4D methods that assume known camera poses or operate on static scenes, this work handles dynamic, multi-object manipulation without pose inputs at test time.
2. Quality: Experiments are thorough, covering both simulation (LBM) and real-world domains, with ablations, multiple baselines (SVD variants, 4D Gaussian), and downstream policy evaluation.
3. Clarity: The problem formulation, method description, and results are presented with exceptional clarity. Figures and tables are informative and well-designed.
4. Significance: Enables pose-free novel-view video generation for robotics—a practical advance for real-world deployment where extrinsic calibration is often unavailable or unreliable. The demonstrated success in policy extraction via off-the-shelf trackers lowers the barrier for applying generative models in robotics.

**Weaknesses:**

1. Computational Cost: Inference takes ~30 seconds per 10-frame rollout (Table 3), which limits real-time or closed-loop use. While acknowledged in §5, more discussion on latency-accuracy trade-offs or potential optimizations (e.g., sparse prediction, distillation) would strengthen practical impact.

2. Real-World Data Scale: The real-world dataset includes only 20 demos per task. While fine-tuning from simulation helps, it’s unclear how performance scales with more diverse real data or more complex scenes (e.g., clutter, deformables).

3. Dependency on FoundationPose: The policy pipeline assumes access to a high-quality 6DoF tracker and a CAD model of the gripper. Generalization to arbitrary tools or unmodeled objects is not explored.

**Questions:**

1. Camera Pose at Training: The method requires camera extrinsics during training to project pointmaps into a shared frame. How sensitive is performance to errors in these extrinsics? Could the system be made fully self-supervised (e.g., via structure-from-motion)?

2. Generalization Beyond Grippers: The pose tracking focuses on the robot end-effector. Can the same 4D video be used to track object poses (e.g., cereal box, apple) for object-centric manipulation? This would broaden applicability.

3. Failure Modes: In Table 2, Task 3 has the lowest success rate (~53%). What are the dominant failure modes? Are they due to video generation errors, pose tracking inaccuracies, or open-loop execution?

---

> ### Author Response · Authors · 2025-11-19
> **Response to reviewer ctzc**
>
> We really appreciate the reviewer’s positive assessment and address the questions raised below.
>
> **Questions:**
> > Camera Pose at Training: The method requires camera extrinsics during training to project pointmaps into a shared frame. How sensitive is performance to errors in these extrinsics?
>
> There is some tolerance to errors in the camera extrinsics, as long as the projected pointmaps from view v_m remain largely aligned with those in the reference view v_n. In our real-world experiments, the cameras are calibrated—but not perfectly accurate compared to the ground-truth extrinsics available in simulation—and our method still performs well, indicating robustness to moderate calibration noise.
>
> > Could the system be made fully self-supervised (e.g., via structure-from-motion)?
>
> Yes, this is a good point! Though not a main focus of the work, the system is possible to be made fully self-supervised. If a 3D map of the scene can be reconstructed—either via classical structure-from-motion or via learning-based methods such as VGGT [1] —it is possible to estimate camera poses directly from RGB images, eliminating the need for camera calibration when collecting the training data in real world.
>
> > Generalization Beyond Grippers: The pose tracking focuses on the robot end-effector. Can the same 4D video be used to track object poses (e.g., cereal box, apple) for object-centric manipulation? This would broaden applicability.
>
> Yes, the same 4D video can be used to track object poses. FoundationPose supports both CAD-based and CAD-free tracking and can estimate the 6-DoF pose for arbitrary objects—not just grippers—given an initial segmentation mask, camera intrinsics, and RGB-D videos. In our paper, we focus on the robot end-effector and we assume its CAD model is usually known. For everyday objects without CAD models, FoundationPose can still operate by capturing a small set of images around the object and reconstructing its geometry, enabling object-centric manipulation as well.
>
> > Failure Modes: In Table 2, Task 3 has the lowest success rate (~53%). What are the dominant failure modes? Are they due to video generation errors, pose tracking inaccuracies, or open-loop execution?
>
> The dominant failure modes in Task 3 are missed grasps—both when lifting the apple from the bowl and when picking it up from the shelf. These failures stem from a combination of factors: occasional generation inaccuracies that lead to subsequent pose-tracking drift, as well as the open-loop execution nature of the framework. Grasping the apple is particularly challenging because the object is small and sometimes partially occluded, making its geometry harder to reconstruct accurately and the gripper–object interaction sometimes inaccurate in the generated frames. Future work may investigate active perception methods in which the system adaptively chooses viewpoints, such as using close-up views to improve grasp accuracy and farther views to better observe the workspace during object transport.
>
> ---
>
> **References:**
>
> [1] Wang, Jianyuan, et al. "Vggt: Visual geometry grounded transformer." Proceedings of the Computer Vision and Pattern Recognition Conference. 2025.

---

### Official Review · Reviewer_QvcZ · 2025-11-01

**Soundness:** 3
**Presentation:** 3
**Contribution:** 2
**Rating:** 6
**Confidence:** 3

**Summary:**

This paper proposes a robot control strategy based on image-conditioned 4D video generation.

A 4D video diffusion model is conditioned on an RGB-D image pair; this generates a pair of pointmap videos (eg, that can be deprojected into a dynamic point cloud). The authors then propose to run FoundationPose-based pose tracking on these generated videos to extract gripper poses, which can then be executed on a robot.

This is similar to prior work in using RGB video for robot control (eg, Dreamitate), but includes extra geometry supervision for the video model.

Experimental results show that this results in robot control policies that generalize better to unseen initial object poses and novel camera viewpoints than methods like Dreamitate and Diffusion Policy.

Overall: the core idea of the paper seems simple but valuable. I'm suggesting a weak accept.

**Strengths:**

I found this to be a  well-written paper, with a clear formulation and convincing empirical results. There's reasonably thorough evaluation across stages: for both 4D generation and for learned robot policies.

I think the robot learning community would benefit from these results; it's a nice existence proof for one way video/world models might be useful for robot learning.

**Weaknesses:**

While the baselines are reasonable, they do feel slightly "set up to fail". For video generation the baselines are RGB only while the proposed method is trained on RGB-D; same for policy rollout. The core message of the paper, however, is in part that the geometry component is critical so perhaps this is fine.

While the approach is elegant and the results are strong, the system depends heavily on multi-view RGB-D data and an external pose tracker (FoundationPose). This raises questions about scalability to real-world deployment, especially in less instrumented or monocular setups.

The reliance on diffusion-based generation also makes inference relatively slow, which (for now) limits applicability to closed-loop control or online replanning.

**Questions:**

How robust is FoundationPose on generated 4D video? Are any of your task failures caused by perception failures, for example if you cannot successfully track the gripper pose or open/close state?

The paper currently only presents task success rates for unseen object poses/novel camera viewpoints. It makes sense to me that methods with less 3D inductive bias would fail here. Do you have success rates for *seen* object poses and/or *fixed* camera viewpoints? It would be helpful, for example, to know how well the 4D video-based approach does compared to RGB video or vanilla diffusion policy when there's less of a train/test gap.

---

> ### Author Response · Authors · 2025-11-19
> **Response to QvcZ**
>
> We would like to thank the reviewer for acknowledging our task and experimental results. Here are the responses to the concerns raised.
>
> > For video generation the baselines are RGB only while the proposed method is trained on RGB-D; same for policy rollout.
>
> For video generation baselines (Table 1, Page 6), [OURS w/o MV attn] takes RGB-D images similarly as our method, but ablates out the multi-view cross attention layers. In addition, though the [4D Gaussian] baseline takes in RGB image, it involves a step to predict depth labels using the DepthAnything model and uses RGB-D data to generate the 4D Gaussian map.
>
> In terms of policy rollout, we actually implement the [DP3] baseline (Table 2, Page 9) which takes in global point clouds projected from RGB-D images as input condition to the policy. Quoted from Line 460-464:
>
> *DP3 (Ze et al., 2024): To compare our method with a behavior cloning method that takes in RGB-D information, we implement DP3, a UNet-based diffusion model that predicts future robot end-effector trajectories conditioned on global 3D point clouds with colors. Following the DP baseline setup, we randomly sample RGB-D images from two training camera views and aggregate their projected global point clouds as input condition. The evaluation setting is the same as the DP baseline.*
>
> **Questions:**
>
> > How robust is FoundationPose on generated 4D video? Are any of your task failures caused by perception failures, for example if you cannot successfully track the gripper pose or open/close state?
>
> Thanks the reviewer for catching this point! We added an additional section in Appendix A.6 (Page 20) to analyze the pose tracking accuracy. Please see the updated manuscript.
>
> In addition, the action extraction method is not limited to FoundationPose—although we find it to be a general and reliable choice in our setting. An alternative is to use an inverse dynamics model (IDM), as in UniPi [1], but this approach requires a substantial number of robot demonstrations covering the task workspace to provide the necessary image-action supervision for training the model.
>
> > Do you have success rates for seen object poses and/or fixed camera viewpoints?
>
> Yes. As shown below, DP3 shows limited improvement, likely because it takes a global pointmap as input and therefore is relatively invariant to viewpoint changes. Diffusion Policy performs better on seen object poses and seen camera views, but it still does not surpass OURS—partly because the dataset contains only 25 demonstrations, whereas DP typically requires around 100 demonstrations to perform stably. Dreamitate also improves under seen conditions, but many of its failure cases arise from tracking errors caused by missing depth information and inaccuracies in the generated video frames.
>
> | Model            | Task 1 | Task 2 | Task 3 | Avg  |
> |------------------|--------|--------|--------|------|
> | Dreamitate       | 0.17   | 0.23   | 0.10   | 0.17 |
> | Diffusion Policy | 0.30   | 0.37   | 0.00   | 0.22 |
> | DP3              | 0.26   | 0.27   | 0.00   | 0.18 |
> | **OURS**         | **0.80** | **0.67** | **0.56** | **0.68** |
>
> ---
> **References:**
>
> [1] Du, Yilun, et al. "Learning universal policies via text-guided video generation." Advances in neural information processing systems 36 (2023): 9156-9172.

---

### Author Response · Authors · 2025-11-26
**Follow-Up on Rebuttal Discussion**

Dear Reviewers,

As the discussion deadline approaches next week, we would like to follow up to see whether you have any remaining concerns or questions that we can address. Thank you again for your constructive feedback on our paper!

Best,
Paper 6362 authors

---

### Meta-Review · Area_Chair_VZS7 · 2026-01-10

**Summary:**

The paper presents a geometry-aware 4D video generation framework for robot manipulation, leveraging cross-view pointmap alignment and pretrained video diffusion models. Reviewers generally recognize the novelty in integrating multi-view geometric supervision with temporal coherence, enabling pose-free novel-view video generation and downstream policy extraction. Experiments on both simulated and real-world tasks demonstrate improvements over strong baselines, particularly in generalization to unseen viewpoints and multi-object scenarios. While some reviewers raised concerns about dataset scale, inference speed, and baseline comparisons, the proposed approach is deemed sound and practically valuable.

**Reviewer Concerns:**

The rebuttal addresses major concerns: baseline comparisons now include RGB-D variants and DP3; pose tracking robustness is clarified with additional analysis; success rates for seen object poses and fixed cameras are reported; differences from prior spatio-temporal methods and sensitivity analyses for cross-view attention are discussed; multi-view scaling and geometric consistency mechanisms are clarified. Remaining minor concerns include inference latency for real-time applications and dependency on multi-view RGB-D datasets, which are acknowledged and discussed as future directions.

**Reviewer Scores:**

Based on the discussion, Reviewer 1 would likely increase their score from 6 to 7, recognizing the clarified baselines and success rate analysis. Reviewer 2, already supportive, would maintain an 8. Reviewer 3 would likely increase from 6 to 7 after the detailed clarification of geometric supervision, cross-view attention, and additional tasks. Reviewer 4, initially at 4, would likely raise to 6, given rebuttal clarifications on reproducibility, baseline justification, and empirical validation, supporting overall acceptance.

---

### Decision · Program_Chairs · 2026-01-26

Accept (Poster)